# Consent to minimally invasive tissue sampling procedures in children in Mozambique: A mixed-methods study

Khátia Munguambe[1,2]*, Maria Maixenchs[1,3], Rui Anselmo[1], John Blevins[4], Jaume Ordi[3,5], Inácio Mandomando[1,6], Robert F. Breiman[4], Quique Bassat[1,3,7,8,9], Clara Menéndez[1,3,9]

1 Centro de Investigação em Saúde de Manhiça (CISM), Cambeve, Maputo, Mozambique, 2 Faculdade de Medicina, Universidade Eduardo Mondlane (UEM), Maputo, Mozambique, 3 Barcelona Institute for Global Health/ Hospital Clínic—Universitat de Barcelona, Barcelona, Spain, 4 Emory Global Health Institute, Emory University, Atlanta, Georgia, United States of America, 5 Department of Pathology, Hospital Clinic-Universitat de Barcelona, Barcelona, Spain, 6 Instituto Nacional de Saúde (INS), Ministério da Saúde, Maputo, Mozambique, 7 Institución Catalana de Investigación y Estudios Avanzados (ICREA), Barcelona, Spain, 8 Pediatric Infectious Diseases Unit, Pediatrics Department, Hospital Sant Joan de Déu (University of Barcelona), Barcelona, Spain, 9 Consorcio de Investigación Biomédica en Red de Epidemiología y Salud Pública (CIBERESP), Madrid, Spain

* khatia.munguambe@manhica.net

**Data Availability Statement:** Data cannot be shared publicly because of ethical considerations particularly the potential of participant identity disclosure. Data are available from the Manhiça

## Abstract

### Background

Minimally invasive tissue sampling (MITS), also named minimally invasive autopsy is a post-mortem method shown to be an acceptable proxy of the complete diagnostic autopsy. MITS improves the knowledge of causes of death (CoD) in resource-limited settings. Its implementation requires understanding the components of acceptability, including facilitators and barriers in real-case scenarios.

### Methods

We undertook a mixed-methods analysis comparing anticipated (hypothetical scenario) and experienced (real-case scenario) acceptability of MITS among relatives of deceased children in Mozambique. Anticipated acceptability information was obtained from 15 interviews with relatives of deceased children. The interview focus was on whether and why they would allow the procedure on their dead child in a hypothetical scenario. Experienced acceptability data were obtained from outcomes of consent requested to relatives of 114 deceased children during MITS implementation, recorded through observations, clinical records abstraction and follow-up informal conversations with health care professionals and semi-structured interviews with relatives.

### Results

Ninety-three percent of relatives indicated that they would hypothetically accept MITS on their deceased child. A key reason was knowing the CoD to take preventive actions; whereas the need to conform with the norm of immediate child burial, the secrecy of

Health Research Centre (CISM), based on the Institutional Data Access Policy. Researchers whose objectives meet the criteria for confidential data access may contact francisco. saute@manhica.net (President of the CISM Scientific Committee) or sozinho.acacio@manhica. net (President of CISM IRB). Data request shall consist of a formal letter addressed to the institution's Scientific Director. Should the response be favourable, researchers can then submit an analytical and publication plan to the Internal Scientific Committee. Should there be clearance, data transfer shall proceed under a Data Transfer Agreement (DTA) between the institution requesting permission for data access and the CISM. Selection of specific data to be shared and removal of potentially identifiable information will be performed prior to data sharing.

**Funding:** This study was funded by the Bill & Melinda Gates Foundation. Specifically, JO received Bill & Melinda Gates Foundation award for part this work: grant reference OPP1067522. URL: https:// www.gatesfoundation.org; RFB received Bill & Melinda Gates Foundation award for part of this work: OPP1126780. URL: https://www. gatesfoundation.org. The funders had no role in study design, data collection and analysis, decision to publish, or preparation of the manuscript. CISM is supported by the Government of Mozambique and the Spanish Agency for International Development (AECID). ISGlobal is a member of the CERCA Programme, Generalitat de Catalunya (http://cerca.cat/en/suma/).

**Competing interests:** The authors have declared that no competing interests exist.

perinatal deaths, the decision-making complexity, the misalignment between MITS' purpose and traditional values, lack of a credible reason to investigate CoD, and the impotency to resuscitate the deceased were identified as potential points of hesitancy for acceptance. The only refusing respondent linked MITS to a perception that sharing results would constitute a breach of confidentiality and the lack of value attached to CoD determination. Experienced acceptability revealed four different components: actual acceptance, health professionals' hesitancy, relatives' hesitancy and actual refusal, which resulted in 82% of approached relatives to agree with MITS and 79% of cases to undergo MITS. Barriers to acceptability included, among others, health professionals' and facilities' unpreparedness to perform MITS, the threat of not burying the child immediately, financial burden of delays, decision-making complexities and misalignment of MITS' objectives with family values.

## Conclusions

MITS showed high anticipated and experienced acceptability driven by the opportunity to prevent further deaths. Anticipated acceptability identified secrecy, confidentiality and complex decision-making processes as barriers, while experienced acceptability revealed family- and health facility-level logistics and practical aspects as barriers. Health-system and logistical impediments must also be considered before MITS implementation. Additionally, the multiple components of acceptability must be taken into account to make it more consistent and transferrable.

## Introduction

A major challenge of the post-2015 Global Development Agenda is the disparity of mortality data availability and quality between high-income and low and middle-income countries (LMICs), due to poor reliability of available tools for cause of death (CoD) assessment in the latter settings. Currently, CoD determination in LMICs is often overlooked or reliant on verbal autopsies, a method subject to a high degree of inconsistencies and misdiagnosis [1,2]. To overcome this problem, minimally invasive autopsy (MIA) methods, lately referred to as minimally invasive tissue sampling (MITS), have been recently developed and validated [3–7] and currently being implemented as a tool for mortality surveillance purposes [8–10].

MITS is a post-mortem procedure consisting of obtaining samples from key organs and body fluids, using biopsy needles, which are subsequently analysed using histopathological and microbiological methods. [5,6,11]. Increasing interest in implementing MITS in LMICs is based mainly on its higher feasibility compared to the complete diagnostic autopsy (CDA) and the need to deploy better methods for CoD investigation to supersede the suboptimal currently utilized ones. A key aspect of the feasibility differential between the two methods is that, with one exception so far studied [12], 7 studies suggest that MITS have potentially higher consent rates compared to the CDA [13–19]. It is assumed that, as MITS hardly affects the physical integrity of the body, the impact on families' state of mind is lower, and healthcare providers' discomfort in interacting with relatives to request a post-mortem examination reduces [16,20,21].

Knowledge about consenting to MITS methods was initially drawn from assessments of acceptability of post-mortem procedures in general, conducted predominantly in high-income countries [13,22–24]. Studies conducted in LMICs, or among immigrant populations residing in high-income countries, suggested that MITS methods might be acceptable even in settings

where post-mortem procedures were assumed to be unfeasible, such as rural, limited-resource areas with no previous community awareness of post-mortem procedures, or among communities with religious values unfavourable to the concept of invasive post-mortem manipulation including the removal of body organs [13,15,19,25]. However, an important limitation of most of these studies is that they rely on hypothetical case scenarios based on respondents´ assessment on whether and why they would agree to MITS, even though they had never experienced it. One study conducted with parents of both dead and living children, had an opportunity to explore acceptability among parents of deceased children who had undergone a MITS, although the vast majority of study participants had not experienced a MITS, therefore the conclusions were drawn on a vast majority of hypothetical understanding of MITS [26]. It is unclear whether the findings obtained from that approach, which captures intentions to accept an intervention, could be extrapolated to actual scenarios during the course of the real-life implementation of the procedure [27]. Several reports indicated that barriers to implementation of interventions on different public health fields were minimized when recommendations drawn from acceptability assessments were followed [28–32]. However, most of these studies did not assess acceptability during the actual intervention, and thus it is unclear how the predicted barriers translated into *de facto* impediments to acceptability.

Acceptability encloses two distinct concepts, namely, *anticipated acceptability*, resulting from assessments prior to the intervention implementation, and *experienced acceptability*, which is assessed during the implementation of the intervention [27]. To our knowledge there are no published studies confronting and questioning the components of each of these two acceptability concepts, the factors contributing to each of them and the transferability of one into the other.

The aim of this study was to compare anticipated versus experienced acceptability of MITS procedure among relatives of deceased children in southern Mozambique, taking into account the specific components of acceptability and factors explaining it, in order to provide evidence for context-specific best practices in the implementation of mortality surveillance systems requiring the post-mortem manipulation of corpses to establish the CoD.

## Methods

### Study site and population

This analysis draws from a continuum of social behavioural assessments nested in two clinical and/or epidemiological studies. The first was the Cause of Death Determination using the Minimally Invasive Autopsy (CaDMIA) study, which aimed to validate MITS in Mozambique and Brazil, with a social behavioural feasibility and acceptability study conducted in Mozambique, Mali, Kenya, Gabon and Pakistan [4–7,33]. The second is the Child Health and Mortality Prevention Surveillance (CHAMPS), which has implemented MITS for mortality surveillance in Mozambique, South Africa, Kenya, Ethiopia, Sierra Leone, Mali and Bangladesh, with a social behavioural assessment in all sites [9,10,26,34,35]. The change in terminology from MIA to MITS occurred before the launch of MITS in the CHAMPS study, in order to avoid negative perceptions linked to the term "autopsy".

In Mozambique, the acceptability component of both studies took place in Manhiça District. The District which comprises a town surrounded by rural areas populated by 183,000 inhabitants. Manhiça is served by a District Hospital (MDH), a Rural Hospital, and 10 peripheral health centres [36]. Since 1996, the population is under the demographic health surveillance system (DHSS) that captures pregnancies, births, deaths and in-and out migrations [37,38]. Additionally, a health facility-based morbidity surveillance system was established to capture data on paediatric inpatient and outpatient visits [39]. In the area, 18% of the population comprises children under the age of five (U5), with a reported mortality rate of 100 per

1000 live births in 2011. According to historical verbal autopsy data, malaria (22%), pneumonia (10%), HIV/AIDS (8%), diarrhoeal diseases (8%) and malnutrition (6%) account for most CoDs in U5 children [36,37].

The study population of this analysis comprises family members (also referred to as relatives) of U5 children (including stillbirths) who died from September 2013 to April 2015 and December 2016 to December 2017.

### Study procedures

Anticipated acceptability variables derived from a secondary analysis of data from the ethnographic component of the CaDMIA study on local attitudes and perceptions related to death and willingness to know the cause of death described in detail elsewhere [15,40]. From September 2013 to April 2015, deaths in the previous 1–30 days within or outside the health facilities (HF) were notified by HF staff, DHSS field workers, or community informants to social behavioural sciences (SBS) research team members, who then invited the closest relative present at the time of death to a one-on-one semi-structured interview. If consent was granted, the interviews took place at the health-facility (waiting area at the morgue, the paediatric ward or the maternity ward) or at the respondent's home or workplace, depending on their preference. The interview followed a guide of open-ended questions encouraging respondents to talk about whether and why they would consider or not the performance of MITS (explained to participants as the use of a semi-automated needle to obtain very small pieces of each organ, without the need to open up the body, and analysing such samples in the lab for CoD investigation) on their deceased child, should MITS be available at the HF (S1 Appendix).

Experienced acceptability variables were drawn from a mixed-methods study on the feasibility to conduct mortality surveillance using MITS by examining factors influencing their acceptability, practicality and implementation [26]. From December 2016 to December 2017, all deaths of U5 children occurring at the District hospital in the previous 24 hours were notified by HF staff to a mortality surveillance staff, who then approached the relatives to request consent to perform MITS on the deceased child. All eligible cases were considered for inclusion in the experienced acceptability assessment. One in every three consecutive cases was selected for observation of the informed consent (IC) process. The non-participant observation technique was conducted by a SBS team member who kept field notes to register actions and reactions of healthcare providers, the project staff and relatives of the deceased child throughout approach to families, consent request, the MITS process and body release. Registration of these events was based on an observation guide (S2 Appendix). If the observer arrived at the enrolment site after the interaction with the relatives was over, the observer conducted a guided informal conversation with the project's staff who had conducted the IC process. Additionally, for all cases, information was extracted from clinical records, which included socio-demographic data, consent outcome and reasons for not performing MITS. A sub-sample of ten families who had refused MITS were sequentially visited at home and invited for a follow-up semi-structured interview to be conducted at home or a location where the respondent would feel comfortable. During the same period, a matched number of relatives who had consented to MITS were interviewed following the same procedure (S2 and S3 Appendices). All interviews were conducted at the respondent's home.

Fig 1 provides a schematic representation of the different study components.

### Data management and analysis

Anticipated acceptability semi-structured interviews with relatives of deceased children were recorded and transcribed verbatim and underwent content analysis. In order to explore the

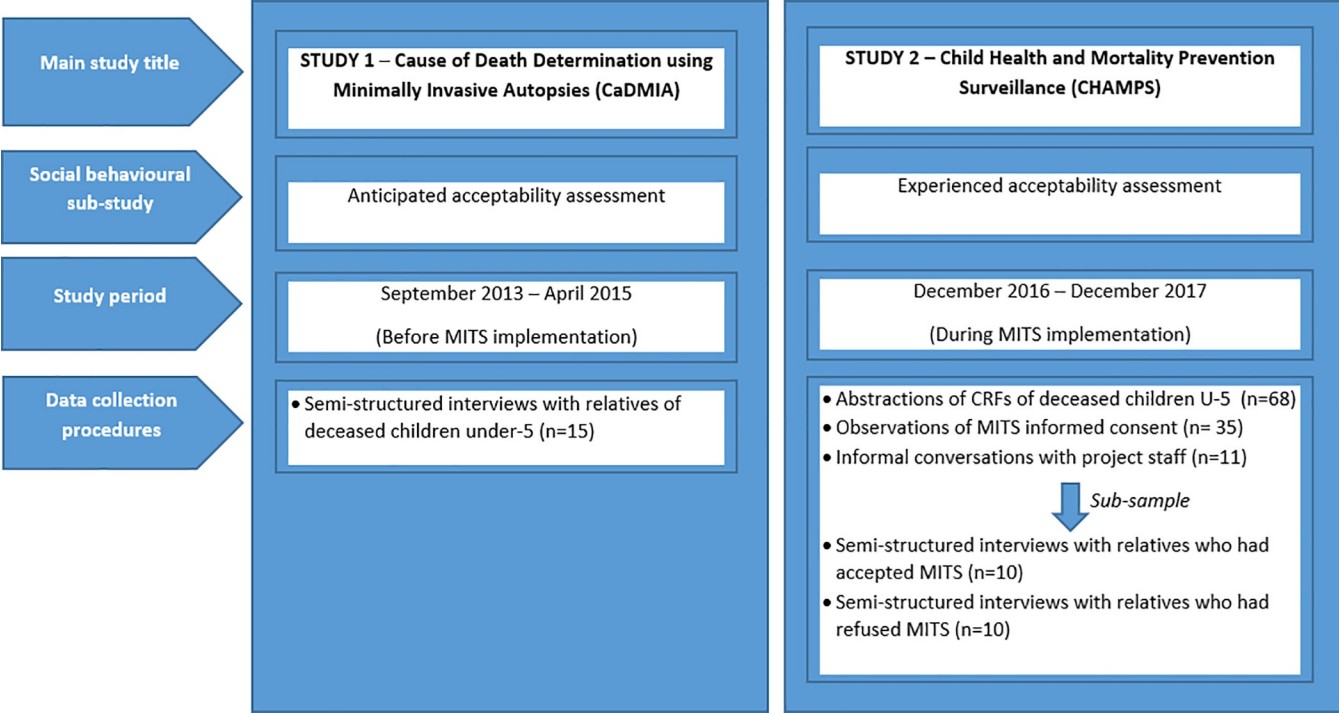

**Fig 1. Schematic representation of the different study components.**

components of anticipated acceptability, information linked to the following five questions was extracted from each transcript: interest in knowing the CoD of their deceased child, willingness to consent to MITS, reasons to accept or refuse it, perceived advantages and disadvantages of executing the procedure, and anticipated barriers and facilitators for MITS future implementation. All extracts of transcription raw data that addressed these questions were fed into a matrix that cross tabulated the contents of what was said by each participant with the headings to each of the above-mentioned questions. This approach generated data-driven categories of responses from the above questions, yielding the specific components of anticipated acceptability and as well as barriers and facilitators of acceptability.

Analysis of the experienced acceptability consisted of triangulating four data sources: direct observation of the IC process; informal conversations with the staff involved in the consent process; eligibility and consent logs; and semi-structured interviews with relatives of deceased children. The observations generated qualitative case-by-case field notes. Eligibility and consent logs provided variables and comments from each case. Interviews with relatives and hospital staff were audio-recorded and transcribed.

The obtained information was linked to the cases in which consent to perform MITS was requested; data were organised in a matrix headed by questions related to experienced acceptability, namely, whether and how the family was approached for consent, expressed interest in the procedure, signed the IC for MITS (and reasons), and if MITS was performed (and reasons, if not). As with the anticipated acceptability analysis, the matrix allowed the generation of emerging categories concerning reasons for consenting and refusals, barriers and facilitators for the execution of the procedure. The main analysis output was qualitative, describing the components of acceptability and indicating how acceptability was expressed, supported by illustrative quotes. When appropriate, this description was accompanied by quantification through frequency counts of cases that contributed to each component of acceptability.

## Ethical considerations

Studies contributing to this analysis received ethical approval from the Manhiça Health Research Centre's IRB (ref: CIBS_CISM10/13), the Clinical Research Ethics Committee of the Hospital Clinic of Barcelona (ref: 2013/8676) and was deemed exempt from further IRB review and approval by the Emory University IRB (ref: IRB00086895). For both anticipated and experienced acceptability interviews, written informed consent was obtained from the family members of the deceased children. During the experienced acceptability study, consent to use observation data for research purposes was part of the overall consent for the CHAMPS study. Relatives who refused MITS did provide consent to be included in all other CHAMPS surveillance activities (and were referred to as "non-MITS cases"). Additional consent was requested to conduct follow-up interviews with selected HF staff and relatives.

## Results

### Participants' characteristics

Of the 35 cases enrolled in the anticipated acceptability study, 15 were relatives of deceased children (4 stillbirths, 4 neonates, 3 infants and 4 children aged 12 to 60 months of age) [15].

During the experienced acceptability assessment, 114 cases of U5 children deaths were relayed to the surveillance team; these cases consisted of 32 stillbirths, 45 early neonatal deaths (up to 7 days old), 3 late neonatal deaths (from 8 to 28 days old), 14 infants between 29 days and 11 months, and 20 children between 12 and 59 months. Overall, 35 cases were directly observed, 11 were recorded through a follow-up informal conversation with MITS consent team once the interaction with the relatives was over, and 68 were registered through case-record forms (CRF) supported by consultations with the MITS consent team.

Additionally, 10 relatives who had refused MITS and an equal number of relatives who had accepted MITS were specifically interviewed.

The characteristics of participants and cases are summarized in Table 1.

### Anticipated acceptability of MITS

Of the 15 relatives of recently deceased children who were asked whether they would hypothetically accept a MITS on their dead child, 14 (93%) replied affirmatively. Only one respondent (the mother of a neonate who had died within 30 to 40 days prior to the interview) kept silent, suggesting a potential refusal, although she freely discussed perceived advantages and disadvantages of the procedure later on in the interview.

When relatives were asked about reasons for MITS acceptance, albeit theoretically (Table 2), one relative considered that a CoD investigation was mandatory. All but one respondent were motivated by their interest in clarifying the CoD. One of the underlying drivers, expressed by the majority, was the strong demand as a parent to know what caused the death, indicated by some resentfulness for not having received explanations from the health facility about their loss. In some cases the driver was the need to answer to funeral-goers who would eventually query on the CoD, which is a common topic of conversation while passing the condolences to the family. Parents felt uncomfortable when unable to provide an answer, especially if hospital help had been sought.

For some relatives MITS was seen as an answer to prevent future disease or death, be it at community level to "save others", as put by one mother, or at family level. Thus, immediate action could be taken to prevent additional fatal events linked to the disease that killed the child in the case of contagious diseases, or to equip mothers with knowledge and skills to protect their future babies from dying, particularly in the case of sudden deaths occurring in apparently healthy babies.

**Table 1. Cases characteristics (family members and deceased children) contributing to the anticipated and experienced acceptability analysis.**

| Characteristics | | n (%) | |
|---|---|---|---|
| | | CaDMIA participants | CHAMPS participants |
| **Family member relationship with child** | Mother | 5 (33) | 36 (32) |
| | Father | 4 (27) | 11 (10) |
| | Mother and father | 0 (0) | 5 (4) |
| | Grandparent* | 6 (34) | 11 (10) |
| | Other combination of members | 0 (0) | 12 (11) |
| | Unknown | 0 (0) | 39 (34) |
| **Family member sex** | Female | 11 (73) | 36 (32) |
| | Male | 4 (27) | 11 (10) |
| | Mixed/ unknown | 0 (0) | 67 (59) |
| **Family member age** | 18–25 | 2 (13) | |
| | 26–59 | 10 (67) | |
| | ≥60 | 3 (20) | |
| | Unknown | 0 (0) | 114 (100) |
| **Family member education level** | No formal schooling | 4 (27) | |
| | Primary | 9 (60) | |
| | Secondary | 2 (13) | |
| | Unknown | 0 (0) | 114 (100) |
| **Religion** | Christian catholic | 2 (13) | |
| | Christian protestant/ evangelic | 6 (40) | |
| | Christian unspecified | 0 (0) | 60 (53) |
| | Muslim | 0 (0) | 1 (1) |
| | Hindu | 0 (0) | 2 (2) |
| | Traditional/ animist | 5 (33) | 26 (23) |
| | Atheist | 1 (7) | 0 (0) |
| | Unknown | 1 (7) | 26 (23) |
| **Child sex** | Female | 8 (53) | 49 (43) |
| | Male | 6 (40) | 65 (57) |
| | Unknown | 1 (7) | 0 (0) |
| **Child age group** | Stillborn | 4 (27) | 32 (28) |
| | Early neonate | 4 (27) | 45 (39) |
| | Late neonate | 0 (0) | 3 (3) |
| | Infant | 3 (20) | 14 (12) |
| | Child | 4 (27) | 20 (18) |
| **Total** | | 15 | 114 |

* Grandparent includes also great-grand parent and great-great grandparent.

In the view of some respondents the procedure might help to address commonly raised speculations about perceived negative actions leading to children's death, such as health professionals' negligence or witchcraft.

Although the majority of relatives were in favour of MITS, participants identified some barriers that would endanger the execution of this approach (Table 3).

Most of such barriers regarded clashes between religious, as well as traditional norms regarding children's deaths and the requisites to perform MITS. There was the concern that MITS could be challenging or even impossible to perform in small children, who must be buried immediately "while the body is still hot". The perceptions of these respondents were

**Table 2. Parents and guardians accounts of anticipated drivers to accept the performance of MITS on their deceased children.**

| Themes and categories | Illustrative quotes/respondent |
|---|---|
| To comply with hospital regulations | *"I will accept because it is a law. . ."*–mother of deceased infant |
| To gain knowledge on the cause of death<br>• Parent's entitlement to know<br>• To overcome the disappointment from not knowing the cause of death<br>• Community's pressure to receive feedback on what caused the death<br>• To help the community with increased knowledge on causes of death | *"The baby had vomits and diarrhoea but nobody informed me about why he died, so I am sad."*–mother of deceased infant<br>*"The mother gave birth at the hospital. Two days later, the baby died at home and the mother returned to the hospital with the dead body. . . People at the community asked about the cause of death, so, if we knew, we would have answered. To know what killed the baby will help the community."*–Grandmother of a deceased new-born<br>*"The MIA should be done as soon as possible after the death, before the family and other people arrive for the ceremonies. . .in that way we could explain to them [the cause of death]"*–grandmother of a stillborn |
| To address suspicion on the cause of death<br>• If negligence is suspected<br>• If witchcraft/traditional cause is suspected<br>• To clarify sudden death | *"I do not know the cause of death, but somebody told me that it was a "traditional illness" and, if the mother is not "traditionally treated", all her babies will die."*–grandmother of a deceased neonate<br>*"I will accept MITS to know the cause of death, to know if the health professionals maltreated or neglected the mother or the baby."*–father of a deceased neonate<br>*"The mother gave birth at the hospital. 2 days later, the baby died at home and the mother came back to the hospital with the dead body. . .to know what killed the baby."*–grandmother of deceased neonate.<br>*"I will accept because my daughter's death was a sudden death and we must know what happened."*–father of a deceased child |
| To prevent further adverse health outcomes<br>• To prevent further deaths in the family<br>• To control contagious diseases<br>• To save lives in the community | *"I will accept MIA because when the cause of the death is known it is possible to prevent it in the future. MIA will allow the family to know why the child died."*–father of deceased child<br>*"If there is a contagious disease, the rest of the family can react."*–father of a deceased neonate<br>*". . .to know the cause of death and to save others."*–mother of deceased infant |

intrinsically linked with the requirement to maintain secrecy about the occurrence of stillbirths and early neonatal deaths. Knowledge of these deaths is restricted to the immediate family and close acquaintances, and female elder relatives who are responsible for the burial ceremonies decide the post-mortem actions to be taken without the involvement of the child´s parents. A few participants mentioned complex decision-making processes that would interfere with the final outcome of consent to MITS. A mother mentioned that although she would favour the performance of MITS on her dead child, her husband, who was absent at the time of the interview, would be the ultimate decision-maker. Other mothers revealed that they had no knowledge regarding what would be decided regarding their child's funeral (or any other post-mortem procedure), leaving that decision up to the church leaders and the child´s grandmothers. One participant noted that priority given to CoD determination over cause of illness investigation among the living was unnatural, therefore a potential barrier for parents to consent. In this regard, a woman went further to say that "not only the results would not bring the child back to life but also they would evoke further pain." Fig 2 illustrates the above findings on factors explaining the willingness to agree to MITS.

## Experienced acceptability of MITS

As shown in Fig 3, out of the 114 eligible CHAMPS cases, the project's staff requested MITS's consent to the families of 101 deceased children. In 13 cases, the relatives were not approached by the project's staff.

**Table 3. Parents and guardians accounts on anticipated barriers to the performance of MITS on their deceased children.**

| Themes and categories | Illustrative quotes |
|---|---|
| Conforming to the norm of burying the child immediately<br>• Requirement to bury a "hot body"–representing a physically and spiritually preserved body<br>• Requirement to bury in "fresh soil"–representing a location that preserves the physical and spiritual integrity | *"The burial must be done while the body is still "hot" and the weather is fresh. . .so early in the morning or at the sun set."*–grandmother of a deceased neonate |
| Secrecy of perinatal deaths<br>• Limited number and specific set of people that should be notified and know about the death<br>• Confining consent and MITS performance to a private location (e.g.: the house of the deceased) | *"It [MITS] should be done at home because it has to be a family secret."*–grandmother of a stillborn<br>*The burial was done less tan 24h after death and no one was informed; just three people, the mother, myself, and a neighbour, took part. To take samples [MITS], the community leader should go with the team in charge of doing it to the family house, without others in the neighbourhood knowing*–grandmother of a deceased neonate<br>*"The baby died in the hospital and the grandmother brought him back home, almost immediately after death, without me seeing the baby and without telling me what would happen next [ceremonies]"*—mother of a deceased neonate |
| Decision making complexity | *"I would accept it [MITS] but it is my husband's decision to accept or not."*–mother of a stillborn |
| No value in investigating the cause of death<br>• Cause of death investigation in tension with the normative of seeking the cause of illness instead of the cause of death<br>• Can cause further agony from recalling the loss<br>• The results will not reverse the death situation<br>• Doubts about the agenda and importance of investigation on cause of death | *"It clashes with tradition because tradition does not look for diseases after death."*–grandmother of a deceased neonate<br>*"To know the cause of death just will make me feel more pain in my heart. . . the child is already dead. What is going to be done with this information?"*–mother of a deceased child |

The most common reason for not approaching relatives was the staff's lack of courage to request consent, given the relatives' verbalized disappointment with the clinical management of the child while alive. The observation team captured some evidence of those cases where relatives had attributed the CoD to the health-facility's mismanagement or neglect leading to staff's discomfort in requesting consent for MITS due to fears of an angry response by relatives (Table 4). Timing between death notification to parents and body release or discarding was also reason for not approaching relatives, as either the families or the health-facility staff took the child's body too quickly while the mortality surveillance team was still organizing the logistics to proceed with the IC and MITS. The third reason captured by observations was the mortality surveillance staff's perception that the mother's state of mind was not suitable to go through the consent process. One of them was unconscious and the other one was in extreme distress. Finally, in one case the study team realised that the child's body was degraded and was therefore not eligible to undergo MITS.

Of the 101 families that were approached, 83 agreed to the procedure on their deceased child (of which 27 were stillbirths, 28 early neonates, 2 late neonates, 10 infants, 16 older children), while 18 families refused it (of which, 12 were early neonates, 1 late neonate, 2 infants and 3 older children). Thus the overall acceptability rate was 82% (100% for stillbirths, 70% for early neonates, 66% for late neonates, 83% for infants and 84% for older children).

The subsample of interviewed family members who had agreed to MITS alluded mainly to their desire to know the CoD, adding that in their opinion MITS seemed to be the only way to

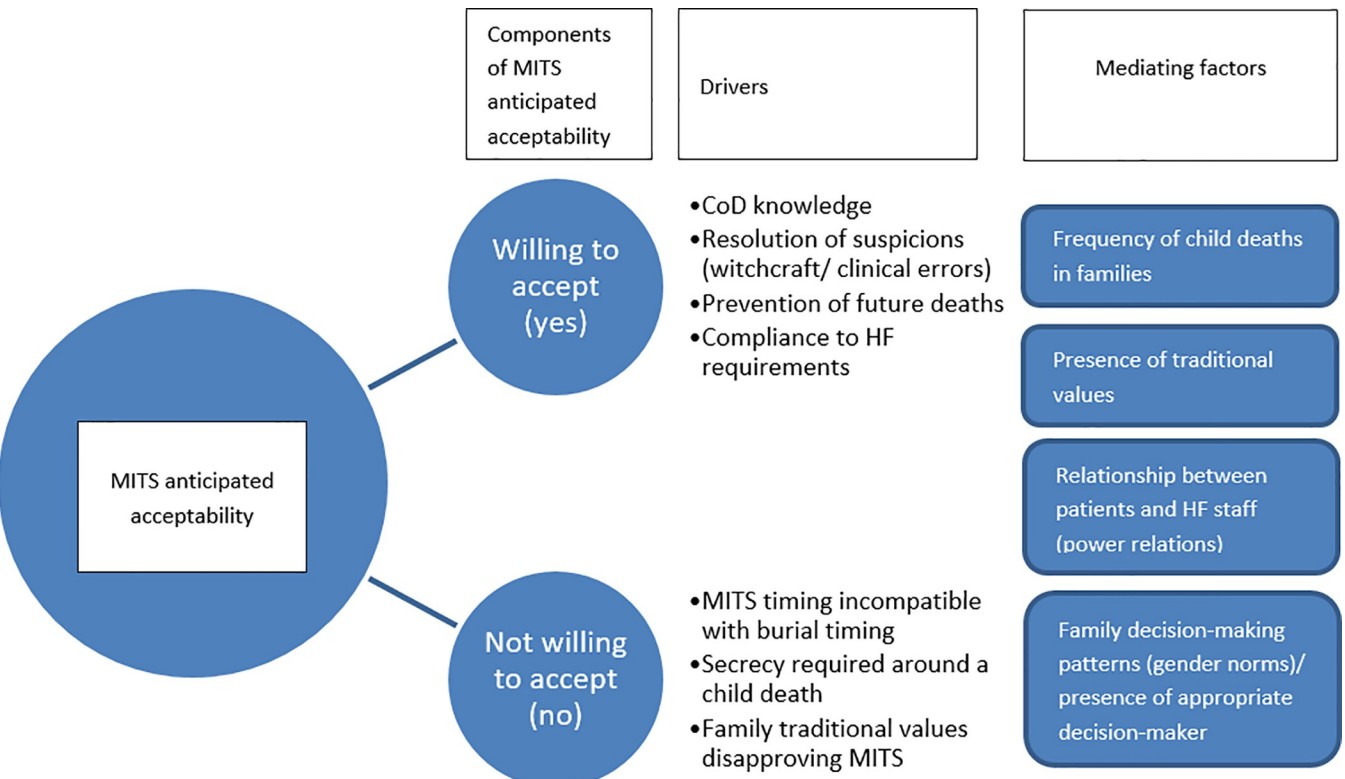

**Fig 2. Dimensions of anticipated acceptance of MITS.** Components, drivers and mediating factors explaining MITS acceptance among relatives of deceased children before the implementation of MITS.

meet their expectations. One parent considered that MITS could help resolve their suspicion of mismanagement of the child while under intensive care. Finally, one family expected that MITS would help end the recurrence of miscarriages or stillbirths in the family. Details of these views are presented in Table 5.

As illustrated by some relatives, an important factor mediating acceptance was the presence of the main decision maker during IC; another facilitator for consent was the fact that they had heard about the intervention before (Fig 4).

Among the 18 families who categorically refused MITS, observations and informal conversations with HF staff revealed that a recurrent reason was the urgency to release the body in time for a burial (Table 6). Undergoing a MITS would potentially lead to postponing the burial to the following day, incurring extra costs to host the mourners, who only depart from the grieving house after the burial. On one hand, the observations revealed the urgency to transport the body while still fresh to enable the caretaker (the grandmother) to piggyback the body as if still alive to be allowed on a public transport van at no extra cost. Otherwise, transporting the body after it had reached post-mortem rigidity would imply arrangements for specific, often unaffordable, transportation services for corpses. On the other hand, complex decision-making processes were involved. For instance, one mother considered the decision had to be taken by the father, although she did not attempt to contact him for reasons unknown to the team. In some cases, although the main care-taker (mostly the mother but in one case a minor, older sister) showed interest in MITS, other relatives (namely fathers, aunts and grandparents) refused the procedure.

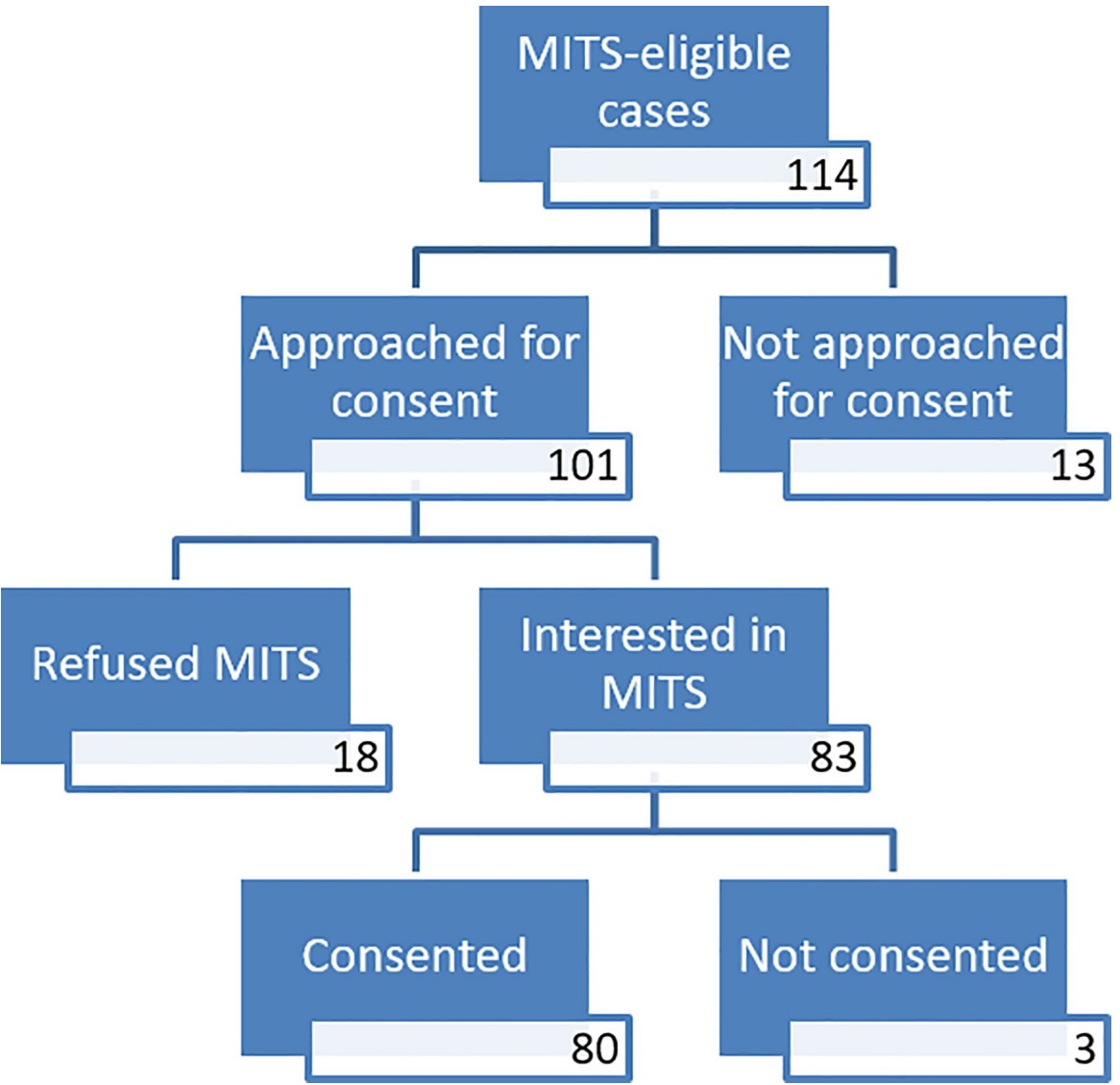

**Fig 3. MITS Consent flow diagram.** Expected sequence of events from informed consent attempt, request and granting or refusal to MITS procedure on deceased children (0–5 Y) in Manhiça District Hospital, Mozambique (2017).

Observations also captured cases of relatives changing their minds after weighing the initially bearable predicted waiting time against the experienced waiting time between the death notification and the outset of MITS performance. In such cases, the relatives who had initially waited ended up giving up because the health-facility was not ready for the procedure to be undertaken as immediately as they initially thought it would be.

In one interview with a father of a deceased child, he explained that he refused the procedure because excessive manipulation of dead bodies, including the removal of body parts, went against his traditional values.

Taking into account the entire cascade, from the identification of eligible cases, followed by missed opportunities to request consent, and also those who during the process changed their minds from acceptance to refusal and finally those who underwent MITS, the proportion of cases in which MITS was performed accounted for 73% of the eligible cases.

**Table 4. Barriers for health professionals to approach family members to request consent to MITS.**

| Themes and categories | Illustrative quotes |
|---|---|
| Underlying tension between family members and health facility staff<br> • Illicit charges impeding potential mutual understanding between relatives and health care workers<br> • Perceived negligence | *"There was no MITS, the family was not even approached by MITS consent taker. There was reticence to do so. . .due to money charges by the health care workers appointed to the maternity ward [the midwife]. In this case the child's grandfather was furious because even though she [the midwife] had received money to take good care of the mother and the baby [informal gratitude], the child ended up dying. I saw no climate for an informed consent to be requested."*–observer's field notes[#] |
| | *"The parents were revolted because their severely sick child could not be referred to Maputo [Maputo Central Hospital] because the ambulance did not have enough oxygen for the child. The health facility staff were trying to say to the parents that the child would be better cared for at MDH because the prognostic was not good at all. . .they explained that the oxygen that was at the ward was the same that the child would be receiving in Maputo and not much else. . .the parents did not care, they think that the child died because of negligence."*–observation field notes[#] |
| Timing<br> • Health professionals' preparedness to engage with families slower than family's readiness to take the body back home | *The family left the hospital while the consent team was getting ready to approach them*–observation field notes |

# Field notes taken during sessions of interactions between project's staff and relatives of deceased children at the MDH.

**Table 5. Reasons for accepting a MITS as reported by parents/caretakers of deceased children (n = 10).**

| Themes and categories | Illustrative quotes |
|---|---|
| To gain knowledge on the cause of death<br> • Trust in modern technology | *"The reason that took us to accept [MITS] is today's times [modernity] because the diseases are plenty, and nowadays the doctors are those who make the right observations and see other things through the machines*–Grandmother of a deceased new born |
| To address suspicion on the cause of death<br> • Possible misconduct by health care professionals | *"We accepted MITS in the hope that the result will reflect our suspicion that the intern nurse gave poisonous medication to the child. . ."*–Father of a deceased infant |
| To prevent further adverse health outcomes<br> • Hope to prevent future miscarriages/stillbirths<br> • To obtain the appropriate cure for child deaths | *The reason why we accepted [MITS] was that the mother is always losing the pregnancies, it is now the second time, all pregnancies with 9 months [gestational age]. The first time she had a stillbirth and the second time she gave birth to a live baby but died straight after. . .we want a healthy child next time*—Grandmother of a deceased new born<br>*What would facilitate the conduction of MITS in my community would be people's drive to know what would have caused the death of the child and desire to obtain the cure so that it does not happen again*–Grandmother of child |
| Involvement of the appropriate decision makers<br> • Alignment between the family members who are present at the time of death and their own role as the principal decision-makers (fathers) | *It wasn't difficult to give consent for MITS because I was with my husband*—Mother of a stillborn<br>*The father of the deceased child did not have any problem in deciding because he was the owner of the baby*–Mother of a stillborn |

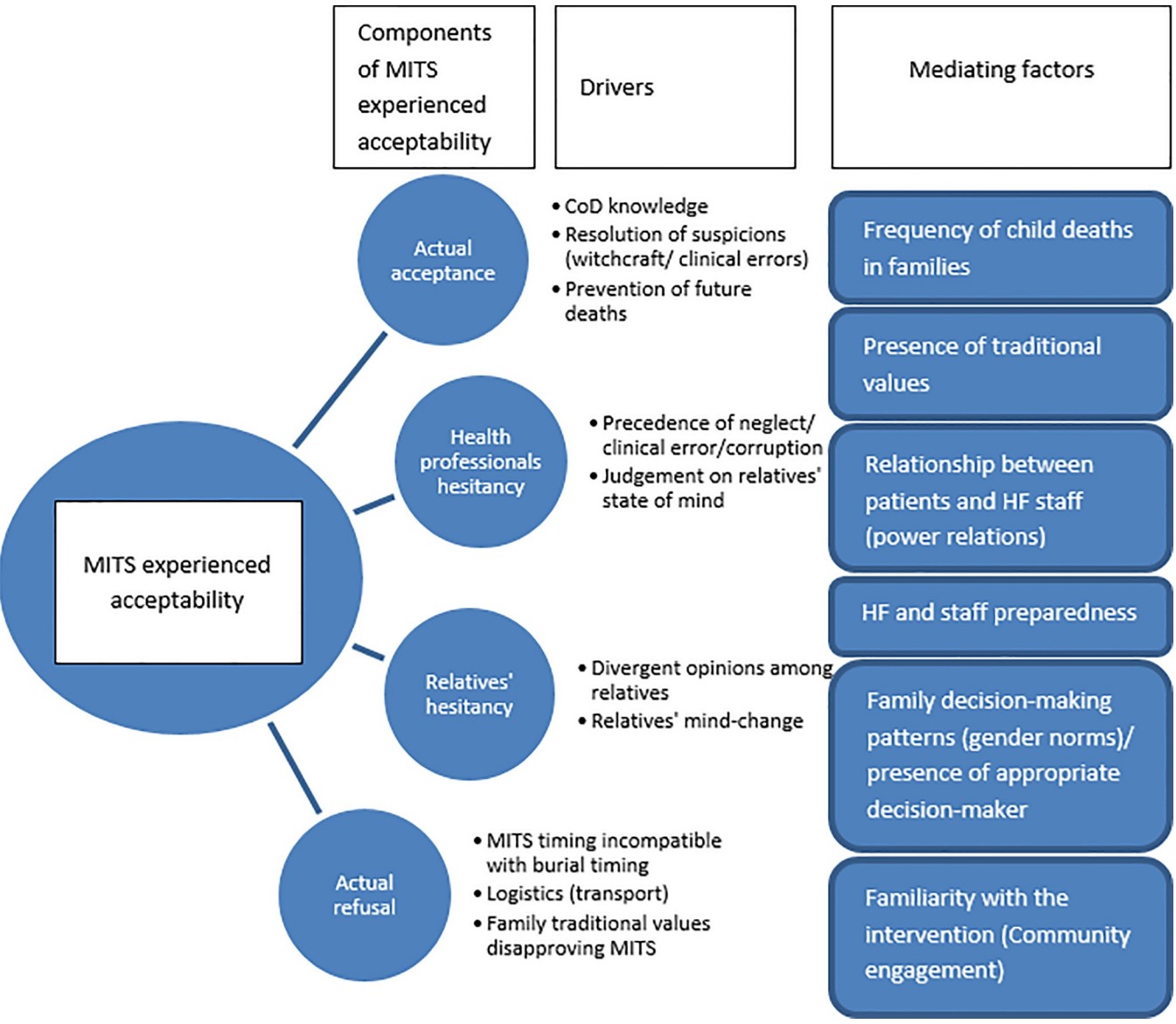

**Fig 4. Dimensions of experienced acceptance of MITS.** Components, drivers and mediating factors explaining MITS acceptance among relatives of deceased children during the implementation of MITS.

## Discussion

To our knowledge, this is the first study that applies the direct observation of experienced acceptability of relatives to perform a MITS on their deceased child in order to triangulate with findings from other classic qualitative enquiry techniques. We found a high level of anticipated acceptability with 93% of relatives willing to accept the performance of the post-mortem procedure on their deceased child when a hypothetical situation was presented to them. Although slightly lower, the level of experienced acceptability was also high (83%). Anticipated acceptability findings agree with studies that assessed willingness to accept MITS in other settings [12,15,19,25].

The small sample size in the anticipated acceptability study (CaDMIA) could also have influenced the relatively lower probability of capturing more hypothetical refusals. This limitation was also considered by authors of an earlier study in Kenya [12].

**Table 6. Reasons for refusing MITS based on direct observation and health staff accounts of refusal cases (n = 18).**

| Themes and categories | Notes from observations and/or informal conversation and quotes from IDIs |
|---|---|
| Decision making complexity | *"The first family member (an older sister of the child) showed interest but they waited for the father who was in South Africa. They ended up taking the body because the father arrived very late in no time to consenting to MITS before the already set up burial time."*–observer's notes |
|  | *"The mother agreed with the procedure, but it was the father, who was martially separated from the mother, who had to provide formal consent. The father, who had moved to Boane district, delegated this mandate to his sister, but she did not feel comfortable to be responsible for the consent, because for that to happen the child should not have been residing with a stepfather. . .so if anything strange was done to the child (including MITS), the child should be taken to Boane district afterwards"*—observer's notes |
|  | *"According to the consent taker, the mother said that she could not decide on anything, and had to wait for the other family members, who took the child's body as soon as they arrived at the hospital."*–field notes on informal conversation with project's staff |
|  | *"The mother explained that she was not able to consent because the father was not there at the hospital. However in practice she took the child immediately back home without waiting for the father."*–observer's notes |
| Conforming with the norm of burying the child immediately | *"I accepted my husband's family´s refusal because there was urgency to bury the child."*- IDI with child's mother |
|  | *"The family had urgency in taking the child back home to comply with the timings for the burial."*–observer's notes |
| Health facility unpreparedness | *"The parents did not deny MITS. They initially showed interest to consent to it. But the MIA was not performed because the morgue table was busy with another body, a case of drowning, and on which the police [forensic department] was running some tests to ascertain the cause of death. Hospital staff explained to me that traumatic deaths were a priority for tests which were run by the police investigators."*–observer's notes |
|  | *"The mother sat down and waited for the next steps, however the consent team members were busy with preparations and took some time to get back to the mother with the paperwork. . .the time they begun the [consent] process, the mother said that it was too late. She was in a hurry to take the foetus for burial."*–observer's notes |
|  | *"The grandmother agreed with the procedure, but it was not done because there were no conditions for that in the morgue (the fridges were full, there were bodies on the table, and there was a bad smell of unclaimed bodies). The municipality had not responded to the hospital's formal request to remove the bodies. After consideration that MITS could not be done on that day, the body was released to the family."*–observer's notes <br> *"There was a failure on the side of the health facility staff, who followed the family's instructions to discard the foetus instead of taking the body to the morgue, not knowing whether the family would eventually consent to a post-mortem procedure. By the time they [MITS team] traced the body it was too late. . .the samples could not be obtained, therefore the family was not formally requested to consent."*–observer's notes |
| Practical transportation requirements | *"The family refused because they wanted to take the body while still fresh to enable to carry [the body] using the public transport, therefore they wanted to take the body immediately."*–CRF abstractions, confirmed by IDI with child's mother |
| Incompatibility with family values | *"The child's father alluded that in their family it is not acceptable the manipulation of the body after the death, including obtaining samples from internal organs."*–observer's notes |
| Unknown reason | *"The family refused and gave no reason."*—observer's notes |

Notwithstanding, in theoretical scenarios acceptability can be an abstract concept, not entirely equivalent to what the actual outcome of consent would be. First, the outcome of an IC process can only be comprehensively assessed in real-case scenarios. Second, consenting

constitutes only one of several components of acceptability; some of the other components were identified when experienced acceptability was examined in this study (e.g., agreeing to the procedure but not being able to comply to it, opposing views among relatives, and health facility staff's hesitance to approach them) [27,41]. These findings add more value to the existing knowledge on the components of acceptability of MITS, which had previously mostly been assessed through questioning relatives of deceased individuals, health professionals and community members in hypothetical case scenarios on the willingness to accept the procedure [15,19].

Recently, there have been attempts to assess the acceptability concept based on theoretical insights, mostly related to health care interventions that focus on treatment [27]. However, due to the life-saving potential of treatment, the knowledge generated by such analyses cannot be transferrable to the understanding of the acceptability of a post-mortem intervention.

The drivers of both anticipated and experienced acceptability were oriented to the expectation that MITS can explain the unknown, ease some suspicions, and reassure relatives with the hope of avoiding further misfortune [41,42]. Importantly, although lightly captured, power relationships between the health care provider and the relatives—which was expressed by the perception that MITS was mandatory—may play a role in acceptability (particularly in anticipated acceptability) [32,42–44]; this may suggest that in the context of consent on deaths occurring in the community, whereby a power shift from the health-facility staff to the relatives and community authorities may be observed, the acceptability to the procedure might be lower.

In alignment with previous studies, the theoretical scenarios highlighted barriers and facilitators to MITS acceptability, which were heavily attached to cultural norms and values (namely, the importance to bury the child as early as possible and under utmost secrecy–especially for neonates, the low value attached to learning about the CoD with no apparent advantage to the family, and incompatible family-level decision making models) [15,19]. The real-case scenarios further revealed practical concerns originating both at family and health-facility level that accounted for the outcome of consent, such as the negative impact of MITS on the timing, financial, and transportation arrangements in preparation for the funeral ceremonies; the high dependence of MITS on health-facility logistics and staff preparedness; and further complexities of the decision-making situation (particularly regarding mother's low decision-making power). The latter factor resonates with results obtained in a CHAMPS site in South Africa [41]. Of note, the comparison between the hypothetical and real case scenarios of the decision-making process regarding the performance of MITS on small children allowed the problem to be viewed from different angles of the family dynamics during the decision making process following the death of a child. Specifically, the anticipated acceptability study highlighted that upon the death of small children (neonates and infants) grandmothers take the lead in the traditional post-mortem events, sometimes not even involving the mother. Drawing from the experienced acceptability study, it was suggested that the addition of an external element (such as a request to perform MITS) to the routine post-mortem events, adds some complexity to the process, which justifies the need not only to involve but more importantly for the final decision to be contingent on the fathers' pronouncement, in their capacity of a legitimate legal representative of the child and the protagonist of power within the family in the context of patriarchal societies [45]. The requirement to sign papers (informed consent), a procedure most grandmothers will not be familiar or comfortable with, adds to this dependency on the child's father.

While the high proportion of consenting relatives supports the suggestion that MITS is a feasible approach to provide cause of death information, it should be noted that it is still an invasive procedure performed on a dead body of a child, which interferes with the above-

discussed sensitive, logistic, traditional and religious requirements and values attached to a death event. Health professionals are not detached from such values either. A previous study on MITS acceptability suggests that, since it is less invasive compared to CDA, MITS might be associated with reduced discomfort for healthcare providers when requesting it to grieving relatives [16]. However, the real-case scenario acceptability analysis revealed important assumptions and reservations from health professionals, which impeded their approaching some of the families. This points to the importance of paying attention to healthcare providers' self-efficacy and disposition to interact with grieving families regardless of how invasive the procedure is. An earlier identified scenario when MITS are not necessarily preferred over CDAs should also be considered with regards to healthcare professionals preparedness to discuss post-mortem examination with families [12].

Of note, the expectations on MITS were overwhelmingly high in this setting, despite the limitation that in a small proportion of cases MITS does not provide conclusive results [4,5,9]. This limitation did not seem to have been fully understood by the relatives. It would be important to discern the extent to which a better awareness of this limitation would influence acceptability. Further studies of relatives' perceptions when they receive inconclusive results would shed more light onto this component of acceptability. The value and meaning attached to the CoD results themselves needs further examination. In this study, while the fear of breaching the confidentiality was captured from the only participant who anticipated a MITS refusal, parents who experienced the procedure expected to discuss results with funeral-goers. This is an unexpected finding, which should be reflected upon with caution especially in case of endemic stigma-prone diseases [15]. This also raises the question on how to manage this expectation over the results, since the funeral happens within days, at the longest, while cause of death results are only delivered months after the death. It is also true that in this setting, besides the vigil and the funeral, the mourning process is marked by a number of remembrance ceremonies, both religious and traditional, spanning across months and even years after the death event. Therefore it is thought that grieving relatives can still digest and discuss the cause of death results that are eventually delivered several weeks to months after the MITS is conducted. However, parents who consented to MITS did not allude to this possibility, probably because they had already experienced the entire MITS process including the lengthy turn-around of results feedback, discarding the overly high expectation of the possibility of feeding back the MITS diagnosis to mourners. This can in turn constitute a barrier to MITS acceptability.

This study highlights that irrespective of the religious background, the timing factor is a significant concern to post-mortem examinations targeting small children in this rural African setting. The relevance of timing is reflected in the observation that even those relatives who had formally consented did change their minds when experiencing delays due to MITS. Issues of timing and secrecy attached to child death events, particularly regarding stillbirths and early neonatal deaths, are critical for implementation of MITS, affecting not only the IC act but also all other MITS related processes, such as case identification and notification, body keeping at and release from health-facilities, transportation and feeding back results to relatives, all of which should be time- and privacy- sensitive.

Lack of health-facility preparedness was an important factor for not accepting MITS. Although this limitation is expected, given the fragile health system, it is likely that relatives may consider this inability to conduct MITS immediately as a lack of respect towards grieving families, and that they perceive it as unaligned with implementers claim on the contribution of MITS to the greater good [46]. Thus, programs that rely on health system infrastructure should contribute to the strengthening and maintenance of such infrastructure so that services can be adequately delivered in parallel to meeting programs' goals. This is especially important for

mortality surveillance platforms, which should be presented as initiatives contributing to mortality reduction.

The study findings also call for the need to continue to build up the knowledge base on the practical experiences of MITS implementation across different contexts. To further advance this knowledge, the recognition that acceptability of a MITS transcends a binary concept captured through relatives' disposition or not to the procedure is crucial. Conduction of qualitative analyses to further deconstruct the concept of MITS acceptability into meaningful and measurable components is needed [47].

These findings can guide the development of more robust instruments to assess MITS acceptability in advance of and during its implementation. For example, the incorporation of more tangible questions that include variables representing practical concerns to all people involved in the procedures, from relatives to health care providers. Furthermore, they may contribute to improve the preparedness of study staff, health facilities, and health care providers to undertake the procedure in the most respectful manner possible to the family.

## Conclusions

There were high levels of relatives' anticipated and experienced acceptability to MITS on their deceased child driven by interest in knowing the CoD. The study identified two components of anticipated acceptability (willingness or not willingness to accept) and four components of experienced acceptability (accepting, inopportunity to voice acceptance due to hesitation by health professionals, relatives changing their minds from accepting to not accepting, and refusing). Secrecy of the event, confidentiality of the results and complex decision-making processes for consent were barriers to anticipated acceptability, while family- and health facility-level logistics and practical aspects were barriers to experienced acceptability. Besides personal, relational, social and cultural barriers to MITS, health-system and logistical impediments should be considered before the procedure's implementation. Studies on anticipated MITS acceptability provide insights that cannot fully inform implementation. Improvement in the tools used for both anticipated and experienced acceptability assessments are needed. Finally, health programs should ensure maximum alignment between their objectives and the individual-, family- and community-level values and priorities; similarly, health-facilities must also be prepared before the implementation of sensitive procedures like MITS, and therefore ethical considerations should include prerequisites for new programs embedded on existing, often weak, health system platforms to make balanced investments between their scientific objectives and the immediate impact on health services quality towards child mortality reduction.

## Supporting information

**S1 Appendix. Semi-structured interview guide used with family members of deceased children (used in study 1).**
(PDF)

**S2 Appendix. Observation guidelines (used in study 2).**
(PDF)

**S3 Appendix. Semi-structured interview guide used with family members who accepted MITS performance on their deceased child (study 2).**
(PDF)

**S4 Appendix. Semi-structured interview guide used with family members who refused MITS performance on their deceased child (study 2).**
(PDF)

## Acknowledgments

The authors would like to thank the families who were engaged in the study, the community leaders and the Manhiça District Hospital's collaborators as gatekeepers for the activities to take place, the observers (Mr. Atanásio Matusse and Ms. Dulce Machava), interviewer (Mrs. Yolanda Uamusse), transcriber (Mr. Elso Ofumane) and the colleagues responsible for data entry and management (Mr. Bento Nhacale and Mr. Arlindo Malheia).

## Author Contributions

**Conceptualization:** Khátia Munguambe, Quique Bassat, Clara Menéndez.

**Data curation:** Khátia Munguambe.

**Formal analysis:** Khátia Munguambe, Maria Maixenchs, Rui Anselmo.

**Funding acquisition:** Jaume Ordi, Robert F. Breiman, Quique Bassat, Clara Menéndez.

**Investigation:** Khátia Munguambe, Maria Maixenchs, Rui Anselmo, John Blevins, Jaume Ordi, Inácio Mandomando, Robert F. Breiman, Quique Bassat, Clara Menéndez.

**Methodology:** Khátia Munguambe, John Blevins, Quique Bassat, Clara Menéndez.

**Project administration:** Maria Maixenchs, Rui Anselmo.

**Supervision:** Khátia Munguambe, Maria Maixenchs, Rui Anselmo, Inácio Mandomando, Robert F. Breiman, Quique Bassat, Clara Menéndez.

**Validation:** Khátia Munguambe.

**Visualization:** Khátia Munguambe.

**Writing – original draft:** Khátia Munguambe.

**Writing – review & editing:** Khátia Munguambe, Maria Maixenchs, Rui Anselmo, John Blevins, Jaume Ordi, Inácio Mandomando, Robert F. Breiman, Quique Bassat, Clara Menéndez.

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
