## [Decision Letter · Decision Letter 0]

10 Sep 2020

PONE-D-20-24297

Consent to minimally invasive tissue sampling procedures in children in Mozambique: a mixed-methods study

PLOS ONE

Dear Dr. Munguambe,

Thank you for submitting your manuscript to PLOS ONE. After careful consideration, we feel that it has merit but does not fully meet PLOS ONE’s publication criteria as it currently stands. Therefore, we invite you to submit a revised version of the manuscript that addresses the points raised during the review process.

We look forward to receiving your revised manuscript.

Kind regards,

Rohina Joshi

Academic Editor

PLOS ONE

Journal Requirements:

"This study was funded by the Bill & Melinda Gates Foundation. CISM is supported by the

Government of Mozambique and the Spanish Agency for International Development

(AECID). ISGlobal is a member of the CERCA Programme, Generalitat de Catalunya

(http://cerca.cat/en/suma/)."

"JO received Bill & Melinda Gates Foundaton award for part this work: grant reference OPP1067522. URL: https://www.gatesfoundation.org

RFB received Bill & Melinda Gates Foundation award for part of this work: OPP1126780. URL: https://www.gatesfoundation.org

The funders had no role in study design, data collection and analysis, decision to publish, or preparation of the manuscript"

4. Please ensure that you refer to Figures 2 and 3 in your text as, if accepted, production will need this reference to link the reader to the figures.

5. Please include your tables as part of your main manuscript and remove the individual files. Please note that supplementary tables (should remain/ be uploaded) as separate "supporting information" files

Reviewers' comments:

Reviewer's Responses to Questions

**Comments to the Author**

1. Is the manuscript technically sound, and do the data support the conclusions?

Reviewer #1: Partly

Reviewer #2: Yes

2. Has the statistical analysis been performed appropriately and rigorously? 

Reviewer #1: I Don't Know

Reviewer #2: Yes

3. Have the authors made all data underlying the findings in their manuscript fully available?

Reviewer #1: No

Reviewer #2: No

4. Is the manuscript presented in an intelligible fashion and written in standard English?

Reviewer #1: No

Reviewer #2: Yes

5. Review Comments to the Author

Reviewer #1: This is an important topic and paper. However, the multiplicity of data sources, time periods, methods, sub-analyses, sub-sub-analyses make it a complete jumble. The paper needs to be much more clearly and better structured before it can be considered for publication. At a very minumum there needs to be some schematic, graphic, or matrix to represent the various study components, the Ns in each, etc.

Reviewer #2: This is a well written mixed method study looking at the acceptability and barriers to MITS in one district of Mozambique. Whilst small in scope, the results and conclusions should help with the planning of interventions involving MITS or other similar mortality surveillance procedures in Mozambique and similar settings. I am not a qualitative researcher, but the methods appear sound based on the variables obtained from the social behavioural assessments from the two MITS studies for ‘anticipated’ and ‘experienced’ acceptability (CaDMIA and CHAMPS). The authors present their results and conclusions well including limitations and the need for further exploration/study both in different sites and expanded to include a more holistic understanding of successful MITS implementation. Given the concerns around the family’s interest in COD in order to be able to tell mourners at the funeral, (and the time-gap between the MITS and COD assignment) I’m interested to know if there were any findings from the ‘experienced acceptability’ arm on this?

Minor issues – Page 14 ‘As illustrated by some relatives, an important factor mediating acceptance was the presence of the main decision maker during IC; another facilitator for consent was the fact that they had heard about the intervention before. Details of these views are presented in table 4.’ – I think this should read ‘table 5’. List of tables appears twice. Figures 2 and 3 are not mentioned in the text. There are several typos in Figures 2 and 3.

It would be good to have the list of the open-ended guided questions as an additional file in case people are interested to see it.

Ethics approvals have been obtained from the relevant authorities.

The authors note that data cannot be shared publicly due to privacy concerns. They provide a contact for data sharing through the Manhiça Health Research centre based on the Institutional Data Access Policy.

6. PLOS authors have the option to publish the peer review history of their article (what does this mean?). If published, this will include your full peer review and any attached files.

Reviewer #1: **Yes: **Philip W Setel

Reviewer #2: No

---

## [Author Response · Author response to Decision Letter 0]

12 Nov 2020

Dear reviewers,

Once again, thank you for reviewing our manuscript entitled “Consent to minimally invasive tissue sampling procedures in children in Mozambique: a mixed-methods study”. We appreciate all the inputs and have made an effort to clarify and address the comments and recommendations raised by the reviewers, in the view that the manuscript will now be suitable for publication in Plos One. 

We hereby present a point by point answer to each of the questions or comments raised by both reviewers as well as the editor. In addition we resubmit the manuscript’s track changed version, the respective clean version, the figures, and the supporting information annexes.

All authors have agreed to the submission of the current version of the manuscript

Respectfully yours, 

Khátia Munguambe (BSc, MSc, PhD)

Centro de Investigação em Saúde de Manhiça (CISM)

Tel/Fax: (+258) 21810002/181

Khatia.munguambe@manhica.net

CP 1929 Manhiça - Moçambique

 

Reviewer #1:

1. “This is an important topic and paper. However, the multiplicity of data sources, time periods, methods, sub-analyses, sub-sub-analyses make it a complete jumble. The paper needs to be much more clearly and better structured before it can be considered for publication. At a very minumum there needs to be some schematic, graphic, or matrix to represent the various study components, the Ns in each, etc.”

Response: Thank you for considering this paper as important. We acknowledge that this is a relevant point and understand that a clarification of the contribution of the different components to the analysis is essential. Therefore we have included a schematic representation of the different study components indicating the designation of the component, the study period, the data collection procedures and/or data sources, and the sample size (number of participants involved) for each data source (referred in lines 186 – 188 of the unmarked version of the revised manuscript).

Reviewer #2: 

1. “This is a well written mixed method study looking at the acceptability and barriers to MITS in one district of Mozambique. Whilst small in scope, the results and conclusions should help with the planning of interventions involving MITS or other similar mortality surveillance procedures in Mozambique and similar settings. I am not a qualitative researcher, but the methods appear sound based on the variables obtained from the social behavioural assessments from the two MITS studies for ‘anticipated’ and ‘experienced’ acceptability (CaDMIA and CHAMPS). The authors present their results and conclusions well including limitations and the need for further exploration/study both in different sites and expanded to include a more holistic understanding of successful MITS implementation. Given the concerns around the family’s interest in COD in order to be able to tell mourners at the funeral, (and the time-gap between the MITS and COD assignment) I’m interested to know if there were any findings from the ‘experienced acceptability’ arm on this?”

Response: We thank the positive appreciation of reviewer #2. The question raised by the reviewer is pertinent and relevant to the continuation of a critical debate on the assessment of cause of death determination acceptability. In response to this question, we added in the discussion (lines 472-480 of the unmarked version of the revised manuscript) a further aspect related to the mismatch between factors influencing anticipated acceptability and experienced acceptability, in that the motivation to accept MITS driven by the possibility of feeding back the MITS diagnosis to mourners was only captured among relatives of deceased children who had never been submitted to the MITS procedure. Most probably the parents who consented to MITS did not allude to this possibility because they had already experienced the entire MITS process including the time lag between MITS performance and results feedback. However, we cannot discard the possibility that this factor might be present, because locally in Mozambique, especially among those who follow Christianity and African Tradition, the mourning and remembrance ceremonies span across months and even years after the death, marked by specific rituals performed at 7 days, 30 days, 6 months, 12 months and once a year thereafter, therefore mourners still have a chance to comment on the cause of death results that are eventually delivered to relatives several weeks to months after the MITS is conducted.

2. “Minor issues – Page 14 ‘As illustrated by some relatives, an important factor mediating acceptance was the presence of the main decision maker during IC; another facilitator for consent was the fact that they had heard about the intervention before. Details of these views are presented in table 4.’ – I think this should read ‘table 5’.”

Response: Thank you for this observation. We corrected the order of the tables, and now we believe it is “Fig 4” (line 360 of the unmarked version of the revised manuscript).

3. “List of tables appears twice.” 

Response: Both lists of tables have been removed, as we learnt from the journal’s guidelines that the tables must appear directly under the paragraphs in which they are cited.

4. “Figures 2 and 3 are not mentioned in the text.”

Response: Thank you for pointing this out. There has been a rearrangement of figures and inclusion of a new figure, as suggested by Reviwer #1 and all figures are now featured in the text.

5. “There are several typos in Figures 2 and 3.”

Response: We appreciate this remark. Typos have been revised and corrected.

6. “It would be good to have the list of the open-ended guided questions as an additional file in case people are interested to see it.”

Response: We agree with the relevance of making the topic guides available. We have therefore incorporated the interview and observation guides as supporting information files, although the majority are in Portuguese and only one is in English (S1-S4 Appendices).

Editor’s comments

1. “Please ensure that your manuscript meets PLOS ONE's style requirements, including those for file naming.”

Response: PLOS ONE’s guidelines and templates were consulted. Adjustments were made accordingly.

2. “Thank you for stating the following in the Acknowledgments Section of your manuscript: "This study was funded by the Bill & Melinda Gates Foundation. CISM is supported by the

Government of Mozambique and the Spanish Agency for International Development

(AECID). ISGlobal is a member of the CERCA Programme, Generalitat de Catalunya

(http://cerca.cat/en/suma/)." We note that you have provided funding information that is not currently declared in your Funding Statement. However, funding information should not appear in the Acknowledgments section or other areas of your manuscript. We will only publish funding information present in the Funding Statement section of the online submission form. Please remove any funding-related text from the manuscript and let us know how you would like to update your Funding Statement. Currently, your Funding Statement reads as follows: "JO received Bill & Melinda Gates Foundaton award for part this work: grant reference OPP1067522. URL: https://www.gatesfoundation.org; RFB received Bill & Melinda Gates Foundation award for part of this work: OPP1126780. URL: https://www.gatesfoundation.org” The funders had no role in study design, data collection and analysis, decision to publish, or preparation of the manuscript"

Response: Thank you for pointing this out. We would like the statement to change to: 

“This study was funded by the Bill & Melinda Gates Foundation. Specifically, JO received Bill & Melinda Gates Foundation award for part this work: grant reference OPP1067522. URL: https://www.gatesfoundation.org; RFB received Bill & Melinda Gates Foundation award for part of this work: OPP1126780. URL: https://www.gatesfoundation.org. The funders had no role in study design, data collection and analysis, decision to publish, or preparation of the manuscript. CISM is supported by the

Government of Mozambique and the Spanish Agency for International Development

(AECID). ISGlobal is a member of the CERCA Programme, Generalitat de Catalunya

(http://cerca.cat/en/suma/)."

3. We note that you have indicated that data from this study are available upon request. PLOS only allows data to be available upon request if there are legal or ethical restrictions on sharing data publicly. For more information on unacceptable data access restrictions, please see http://journals.plos.org/plosone/s/data-availability#loc-unacceptable-data-access-restrictions. In your revised cover letter, please address the following prompts:

Response: There are ethical restrictions on sharing a de-identified data set because these data are on deceased children and contain potentially sensitive information. Even if the data is de-identified, there is a combination of variables such as age, date and place of death, neighbourhood of origin, and several variables related to the child’s mother, to name just a few, which have the potential to make the case identifiable. On top of these concerns, the participant information sheet, which was approved by the local Institutional Review Board and the National Bioethics Committee, was clear to the participants that the data would only be disseminated widely in its aggregated form. Therefore, depending on the objectives of the person or entity requiring the data, data might or might not be granted upon request for the reasons stated above. We hereby provide the relevant institutional contact details for this matter: Dr Sozinho Acácio (sozinho.acacio@manhica.net) – President of the Manhiça Health Research Centre’s IRB; Dr Francisco Saúte (francisco.saute@manhica.net) – President of the Manhiça Health Research Centre’s Scientific Committee.

4. Please ensure that you refer to Figures 2 and 3 in your text as, if accepted, production will need this reference to link the reader to the figures.

Response: Thank you for pointing this out. We have now incorporated all the figures in the text. Please also note that one figure was added, following the suggestion of Reviewer #1.

5. Please include your tables as part of your main manuscript and remove the individual files. Please note that supplementary tables (should remain/ be uploaded) as separate "supporting information" files.

Response: Tables have been included as part of the main manuscripts and individual files were removed.

---

## [Decision Letter · Decision Letter 1]

26 Jan 2021

PONE-D-20-24297R1

Consent to minimally invasive tissue sampling procedures in children in Mozambique: a mixed-methods study

PLOS ONE

Dear Dr. Khátia Rebeca Munguambe,

Thank you for submitting your manuscript to PLOS ONE. Now I have received the reports from all the reviewer's. I have also read the manuscript with high interest.  After careful consideration, we feel that it has merit but does not fully meet PLOS ONE’s publication criteria as it currently stands. Therefore, we invite you to submit a revised version of the manuscript that addresses the points raised during the review process. The major area of concern of the current version of manuscript is the analysis and presentation of qualitative data. The Tables 2-6 have presented qualitative findings. These data have been interpreted using quantitative lens. On the other hand, we conduct research using qualitative approach to know the issue with more in-depth insight; not the numbers. However, these tables have provided themes related to drivers, barriers, etc. with a single quote for each of the themes. This approach does not provide in-depth insight of the problem. Thus, it is required to reanalyze the data using qualitative lens and presenting the data following standard style of qualitative data. 

We look forward to receiving your revised manuscript.

Kind regards,

Mohammad Bellal Hossain, PhD

Academic Editor

PLOS ONE

Reviewers' comments:

Reviewer's Responses to Questions

**Comments to the Author**

1. If the authors have adequately addressed your comments raised in a previous round of review and you feel that this manuscript is now acceptable for publication, you may indicate that here to bypass the “Comments to the Author” section, enter your conflict of interest statement in the “Confidential to Editor” section, and submit your "Accept" recommendation.

Reviewer #1: All comments have been addressed

Reviewer #2: All comments have been addressed

Reviewer #3: (No Response)

2. Is the manuscript technically sound, and do the data support the conclusions?

Reviewer #1: Yes

Reviewer #2: Yes

Reviewer #3: Yes

3. Has the statistical analysis been performed appropriately and rigorously? 

Reviewer #1: I Don't Know

Reviewer #2: Yes

Reviewer #3: Yes

4. Have the authors made all data underlying the findings in their manuscript fully available?

Reviewer #1: Yes

Reviewer #2: No

Reviewer #3: Yes

5. Is the manuscript presented in an intelligible fashion and written in standard English?

Reviewer #1: Yes

Reviewer #2: Yes

Reviewer #3: Yes

6. Review Comments to the Author

Reviewer #1: My concnerns have beeen addressed, as I believe have most of the other comments from other reviewers. I think this revision is improved and worthy of publication.

Reviewer #2: (No Response)

Reviewer #3: This is a good paper with good scientific rigour balancing both qualitative and quantitative aspects of research into MITS. I am impressed with the discussion and conclusion sections which have focused on widening the implications of their study on other studies in other contexts while still maintaining socio-cultural differences that would be there dependent on contexts. The methodology is also well discussed and is comprehensive. An important observation that I have made and would wish it were cleared even by a line is that in anticipatory acceptability, there is a mention of grandmothers as the key decision makers yet in actual acceptability there is a reference to fathers; I would love to have a sentence explaining the differences here and how they came about. A few edits that have to be made:

Line 101, page 5: Not clear ‘who had experienced a MITS’?

Line 114, page 5: Indicate that there are no ‘published’ studies, there are some studies (e.g the Malawian study) only that it is yet to publish

Line 182, page 8: were invited into the hospital for the interview? It is not clear yet it is an important part

Line 476, page 27: not clear on socialisation and results of MITS (grammar)

7. PLOS authors have the option to publish the peer review history of their article (what does this mean?). If published, this will include your full peer review and any attached files.

Reviewer #1: No

Reviewer #2: No

Reviewer #3: **Yes: **Dave Mankhokwe Namusanya

---

## [Author Response · Author response to Decision Letter 1]

16 Jun 2021

Responses to the Comments to the Author

• Reviewer #1 comment: 

My concerns have been addressed, as I believe have most of the other comments from other reviewers. I think this revision is improved and worthy of publication.

Response: We thank reviewer #1 for this very encouraging comment, and for the valuable, constructive comments given on the previous version of the manuscript. We appreciate the consideration that the manuscript is worthy of publication.

• Reviewer #2 comment: 

No Response

Response: We believe that the lack of comments implies an agreement with the new version of the manuscript.

• Reviewer #3 comments: 

This is a good paper with good scientific rigour balancing both qualitative and quantitative aspects of research into MITS. I am impressed with the discussion and conclusion sections which have focused on widening the implications of their study on other studies in other contexts while still maintaining socio-cultural differences that would be there dependent on contexts. The methodology is also well discussed and is comprehensive. An important observation that I have made and would wish it were cleared even by a line is that in anticipatory acceptability, there is a mention of grandmothers as the key decision makers yet in actual acceptability there is a reference to fathers; I would love to have a sentence explaining the differences here and how they came about. A few edits that have to be made:

Response: We are grateful for reviewer#3 appraisal, which appreciates the mixed methods approach to our analysis. 

We believe that the point raised with regards to the apparent shift in the decision-making role of grandmothers between the anticipated and the experienced acceptability assessment is relevant. While the anticipated acceptability study suggested that elder members of the family, notably grandmothers, had an important role in determining the ceremonial procedures, sometimes without the involvement of the deceased children’s mothers, direct observation and interview with relatives of children in the experienced acceptability study revealed the strong presence of fathers in the decision-making process as a new element. This is not to say that grandmothers lose their place in the process, as some quotes reveal their involvement in critical choices resulting from upfront decisions such as the one detailed below: 

“For two cases, there was urgency to transport the body while still fresh to enable the caretaker (in both cases the grandmother) to piggyback the body as if still alive to be allowed on a public transport van at no extra cost. Otherwise, transporting the body after it had reached post-mortem rigidity would imply arrangements for specific, often unaffordable, transportation services for corpses.” - non-participant observer’s notes

There are further illustrations of grandmothers involved in the collective decision to accept the MITS, in real scenarios of MITS implementation, as shown in the quotes below:

 “The reason why we accepted [MITS] was that the mother is always losing the pregnancies, it is now the second time, all pregnancies with 9 months [gestational age]. The first time she had a still birth and the second time she gave birth to a live baby but died straight after…we want a healthy child next time.” - Grandmother of a deceased new born

“The reason that took us to accept [MITS] is today’s times [modernity] because the diseases are plenty, and nowadays the doctors are those who make the right observations and see other things through the machines.” – Grandmother of a deceased new born

What we can infer from the data is that the comparison between the hypothetical and real case scenarios of the decision-making process regarding the performance of MITS on small children allowed the problem to be viewed from different angles of the family dynamics during the decision making process following the death of a child. Specifically, the anticipated acceptability study highlighted that upon the death of small children (neonates and infants) grandmothers take the lead in the traditional post-mortem events, sometimes not even involving the mother. Drawing from the experienced acceptability study, it was suggested that the addition of an external component (such as a request to perform MITS) to the routine post-mortem events, adds some complexity to the process, which justifies the need not only to involve but more importantly for the final decision to be contingent on the fathers’ pronouncement, in their capacity of a legitimate legal representative of the child and the protagonist of power within the family in the context of patriarchal societies. To address this suggestion from reviewer #3, we have revisited the discussion to add the above reflection (lines 444 – 455). 

Line 101, page 5: Not clear ‘who had experienced a MITS’?

Response: We changed the sentence so that it is clearer that the MITS had been performed only on some of the deceased children: “One study conducted with parents of both dead and living children, had an opportunity to explore acceptability among parents of deceased children who had undergone a MITS, although the vast majority of study participants had not experienced a MITS, therefore the conclusions were drawn on a vast majority of hypothetical understanding of MITS” (lines 100 – 104)

Line 114, page 5: Indicate that there are no ‘published’ studies, there are some studies (e.g the Malawian study) only that it is yet to publish

We corrected this passage of text to highlight the lack of published studies, rather than the lack of studies, that specifically compare hypothetical vs experience acceptability. We appreciate the alert on the then upcoming publication of the Malawian study. However, at the time of the submission, we would not have known or have been able to cite an unpublished piece of work. We revisited the literature and are aware of an article (Lawerence et al, 2021) that also focuses on hypothetical acceptability, still not addressing experienced acceptability. We have now updated the introduction (line 115) to highlight the word “published” and the discussion (lines 451 and 454) to include the Malawian study.

Line 182, page 8: were invited into the hospital for the interview? It is not clear yet it is an important part

We revised the sentence and made it clear that the interview would be conducted at home or at a place that was not the hospital (184 – 186).

Line 476, page 27: not clear on socialisation and results of MITS (grammar)

This was changed to “digest and discuss” for more clarity (line 499).

• Editor’s comment:

The major area of concern of the current version of manuscript is the analysis and presentation of qualitative data. The Tables 2-6 have presented qualitative findings. These data have been interpreted using quantitative lens. On the other hand, we conduct research using qualitative approach to know the issue with more in-depth insight; not the numbers. However, these tables have provided themes related to drivers, barriers, etc. with a single quote for each of the themes. This approach does not provide in-depth insight of the problem. Thus, it is required to reanalyze the data using qualitative lens and presenting the data following standard style of qualitative data.

Response: This is the second round of revisions and we are surprised that the concern on the data analysis approach is only raised at this point. We disagree with the position that the analysis approach is not suitable to the purpose of the article, as it was clearly stated in the title and other passages of the text that this was a mixed methods approach. In fact, reviewer #3, whose comments were highly appreciated by the authors, has stated that “This is a good paper with good scientific rigour balancing both qualitative and quantitative aspects of research into MITS”. 

Moreover, we would like to reinforce that we opted for content analysis, rather than thematic analysis, due to the type of data that we had at hand, which in the majority of cases was generated by semi-structured data collection tools, which as the editor knows, do not generate extremely in-depth accounts. In order to minimize the limitations offered by data generated by semi-structured data collection tools, triangulation which we made the most of, allowed to maximize the integrity of the data. In this respect, we are confident that using data from observations and semi-structured interviews, with a good mix of both target groups and different time points relative to the performance of MITS, did allow for such triangulation.

The entire interpretation and conclusions of the paper revolved around the above-mentioned choice of the content analysis approach. Therefore, we trust that reanalysing the data would change the entire structure, conceptual models, and even the study design (which is clearly unfeasible at this point) as this is a secondary data analysis. We understand that what the editor would like us to have done is a thematic analysis, which was not the intention of the investigators from the on-set of the decision to pursue this exercise, which we believe is valuable despite the limitations.

Content analysis is a relevant and entirely valid qualitative data analysis approach. As all other approaches, it has its strengths and limitations. This is far from regarding this approach as invalid. Content analysis can be deductive, rather than inductive, and to do so it is often based on a categorization matrix . Hence, perhaps, the editor’s misconception that qualitative elements have been transformed into quantitative ones. Nonetheless, in order to address this particular concern, we have decreased the allusion to numbers or percentages along the entire text (as it can be seen in the new version of the manuscript). However, there were objectives that required a quantitative-based answer which we strongly feel should be left quantified. Such is the case of proportion of individuals who have accepted or refused the MITS, and the description of participants’ characteristics.

We also do not agree that table 2-6 have been analysed with quantitative lenses, because we were not based on frequency distributions nor on any statistical approach to reach our interpretation. Rather, we made a careful reading of observation reports and interview transcripts to identify the themes linked to the predetermined constructs of acceptability. Additionally, the editor states that each table has a single quote to illustrate each theme. The use of quotes, particularly how and when to use them, is a current and evolving debate in qualitative research. The editor’s point of view is just one of the many current positions. Among this debate, there is the criticism on “papers that read like a laundry list of quotes rather than a story about what the writer learned” . Our approach to the quotes was two-fold: (i) choosing quotes that more thoroughly captured the different themes identified in the data, while (ii) being sensitive to the word limits in most biomedical papers. Notwithstanding, there are numerous instances in which we used more than one quote to illustrate a single theme, contrary to the editor’s remarks that we used single quotes for each theme. Notably: table 2 has 3 quotes for theme # 2 (with two subthemes); 4 quotes to illustrate theme #3 (with 3 sub-themes); and 2 quotes for theme #4 (with 3 subthemes). Table 3 has 3 quotes for theme #2 (2 subthemes); and 2 quotes for theme # 4 (3 subthemes). Table 5 has 2 quotes for each of themes #3 and #4. Finally table 6 has 5 quotes for theme #1, and two quotes for each of themes #3, #4, and #5 with interesting thematic overlaps between some of the quotes. In order to better capture such overlaps, we rearranged the format of table 6, which perhaps will address some of the editor’s concerns.

The reason why we created tables in order to synthesize the data was precisely not to overload the text with the quotes. This is just a choice of presentation that has been seen in many peer-reviewed qualitative research manuscripts . Therefore, with all due respect to the editor’s suggestion, we would like to preserve this style, as it is faithful to the original material generated by the content analysis matrices. At the same time it aligns effortlessly with the research objectives and the conceptual models that were created out of the analysis and enables the audience (which we believe are mostly public health specialists rather than sociologists or anthropologists) to understand the results. However, if the editor feels strongly against the tabulations, we will be open to revert this style to include the quotes intercalating the text, as a classic qualitative paper would look like.

Finally, in the interest of an open discussion, we would appreciate that the three reviewers originally assigned to this manuscript read both the editor’s comment and our response to this issue, so that a consensus is reached regarding the relevance and validity of this manuscript and weather it is suitable for publication as it stands.

---

## [Decision Letter · Decision Letter 2]

25 Oct 2021

Consent to minimally invasive tissue sampling procedures in children in Mozambique: a mixed-methods study

PONE-D-20-24297R2

Dear Dr. Munguambe,

We’re pleased to inform you that your manuscript has been judged scientifically suitable for publication and will be formally accepted for publication once it meets all outstanding technical requirements.

Kind regards,

Mohammad Bellal Hossain

Academic Editor

PLOS ONE

Additional Editor Comments (optional):

Reviewers' comments:

Reviewer's Responses to Questions

**Comments to the Author**

1. If the authors have adequately addressed your comments raised in a previous round of review and you feel that this manuscript is now acceptable for publication, you may indicate that here to bypass the “Comments to the Author” section, enter your conflict of interest statement in the “Confidential to Editor” section, and submit your "Accept" recommendation.

Reviewer #2: All comments have been addressed

Reviewer #3: All comments have been addressed

2. Is the manuscript technically sound, and do the data support the conclusions?

Reviewer #2: Yes

Reviewer #3: Yes

3. Has the statistical analysis been performed appropriately and rigorously? 

Reviewer #2: Yes

Reviewer #3: N/A

4. Have the authors made all data underlying the findings in their manuscript fully available?

Reviewer #2: No

Reviewer #3: Yes

5. Is the manuscript presented in an intelligible fashion and written in standard English?

Reviewer #2: Yes

Reviewer #3: Yes

6. Review Comments to the Author

Reviewer #2: This is the third time I have seen this manuscript. I was satisfied that all my comments and concerns were addressed in the previous version. I see that there are some additional comments from a new reviewer that the authors have now addressed. I agree with the author that methods and presentation of results are appropriate for this public health audience. The authors highlight both the implications of their findings and limitations of their study and suggestions for further investigation. I think this paper should now be published.

Reviewer #3: The article has addressed all the issues I raised. They have also provided comprehensive feedback on any lingering questions.

7. PLOS authors have the option to publish the peer review history of their article (what does this mean?). If published, this will include your full peer review and any attached files.

Reviewer #2: No

Reviewer #3: No

---

## [Editor Report · Acceptance letter]

29 Oct 2021

PONE-D-20-24297R2 

*Consent to minimally invasive tissue sampling procedures in children in Mozambique: a mixed-methods study*

Dear Dr. Munguambe:

I'm pleased to inform you that your manuscript has been deemed suitable for publication in PLOS ONE. Congratulations! Your manuscript is now with our production department. 

Kind regards, 

on behalf of

Dr. Mohammad Bellal Hossain 

Academic Editor

PLOS ONE